# Radiotherapy and Testicular Function: A Comprehensive Review of the Radiation-Induced Effects with an Emphasis on Spermatogenesis

**DOI:** 10.3390/biomedicines12071492

**Published:** 2024-07-05

**Authors:** Ioannis Georgakopoulos, Vassilios Kouloulias, Georgios-Nikiforos Ntoumas, Dimitra Desse, Ioannis Koukourakis, Andromachi Kougioumtzopoulou, George Kanakis, Anna Zygogianni

**Affiliations:** 1Radiation Oncology Unit, 1st Department of Radiology, Medical School, Aretaieion Hospital, National and Kapodistrian University of Athens, Vas. Sofias 76, 115 28 Athens, Greece; geontoumas@yahoo.com (G.-N.N.); koukourioannis@gmail.com (I.K.); annazygo1@yahoo.gr (A.Z.); 2Radiotherapy Unit, 2nd Department of Radiology, Medical School, National and Kapodistrian University of Athens, Rimini 1, 124 62 Athens, Greece; vkouloul@ece.ntua.gr (V.K.); andromachi.kou@gmail.com (A.K.); 3Department of Endocrinology, Athens Naval & VA Hospital, 115 21 Athens, Greece; geokan@endo.gr; 4Unit of Reproductive Endocrinology, First Department of Obstetrics and Gynaecology, Medical School, Aristotle University of Thessaloniki, 541 24 Thessaloniki, Greece

**Keywords:** radiation, radiotherapy, testicular function, spermatogenesis, male fertility

## Abstract

This comprehensive review explores the existing literature on the effects of radiotherapy on testicular function, focusing mainly on spermatogenic effects, but also with a brief report on endocrine abnormalities. Data from animal experiments as well as results on humans either from clinical studies or from accidental radiation exposure are included to demonstrate a complete perspective on the level of vulnerability of the testes and their various cellular components to irradiation. Even relatively low doses of radiation, produced either from direct testicular irradiation or more commonly from scattered doses, may often lead to detrimental effects on sperm count and quality. Leydig cells are more radioresistant; however, they can still be influenced by the doses used in clinical practice. The potential resultant fertility complications of cancer radiotherapy should be always discussed with the patient before treatment initiation, and all available and appropriate fertility preservation measures should be taken to ensure the future reproductive potential of the patient. The topic of potential hereditary effects of germ cell irradiation remains a controversial field with ethical implications, requiring future research.

## 1. Introduction

The term “spermatogenesis” refers to the uninterrupted process of mature sperm cell production, which occurs in the seminiferous tubules of the testes of post-pubescent males. Spermatogenesis is one of the most mitotically active procedures in the human organism, as spermatogonia, the diploid precursors of haploid sperm cells, are constantly undergoing mitotic division, followed by meiosis to produce round spermatids, which then differentiate into mature sperm cells [1]. The continuous cycles of mitotic and meiotic cell division render spermatogenesis quite susceptible not only to incidental errors, intrinsic to the process of DNA replication among others, but also to environmental toxic factors, including chemotherapeutical agents and radiation exposure [1,2,3,4].

Ionizing radiation, used in the context of tumor radiotherapy, causes cellular death, mainly through direct or indirect DNA damage in the nucleus, resulting in apoptosis, mitotic catastrophe, necrosis, senescence and autophagy [5]. One of the basic principles of radiobiology is the law of Bergonié–Tribondeau, which states that the radiosensitivity of a tissue is proportional to the proliferation rate of its cells and inversely proportional to the degree of cell differentiation [6]. Of note, the above statement was based on observations made by irradiating rat testicles and studying the effects on spermatogenesis, indicating the association between radiation exposure and its impact on normal testicular function as early as at the beginning of the previous century [7]. The aim of this comprehensive review is to summarize the existing evidence on the effects of radiotherapy primarily on the process of spermatogenesis and secondarily on endocrine testicular function.

## 2. Overview of the Process of Human Spermatogenesis

Figure 1 describes the process of human spermatogenesis and correlates the various cellular components of the testes with the corresponding minimum radiation dose that results in their dysfunction and/or death. During embryogenesis, primordial germ cells migrate to the developing testes and give rise to testicular germ cells, called spermatogonia [8]. Spermatogonia are organized into two to three layers and occupy the outer portion of the seminiferous tubule epithelium, immediately adjacent to the basement membrane [9]. Leydig cells are located in the interstitial connective tissue in between the seminiferous tubules, and under the regulation mainly of Luteinizing Hormone (LH), they are the primary source of testosterone and other androgens in post-pubescent males [10]. Sertoli cells are the second component of the seminiferous epithelium and serve to facilitate and support the division and differentiation of spermatogonia into mature spermatozoa under the influence of Follicle-Stimulating Hormone (FSH) and testosterone [11]. Spermatogenesis starts with the onset of puberty at a mean age of 13 years old and continues uninterrupted throughout the whole lifespan. The initial step is the mitotic division and differentiation of spermatogonia into primary spermatocytes. Each of the primary spermatocytes undergoes a first meiotic division to produce two secondary spermatocytes, which in turn proceed to the second meiosis to produce round spermatids. Finally, spermatids undergo maturation from a round morphology to the characteristic elongated head-and-tail shape to produce mature spermatozoa, which are then released into the tubule lumen. The whole process of spermatogenesis, from the spermatogonium to the mature spermatozoon, lasts approximately 74 days, and the various steps happen in the direction from the basement membrane towards the lumen. Further elaborating on the features of human spermatogonia, there are two types of spermatogonia, type A, further subdivided into A pale (Ap) and A dense (Ad), and type B. Type Ad spermatogonia normally comprise inactive reserve stem cells that only undergo division and transformation into Ap cells as an aftermath of irradiation and cytotoxic treatment. Ap spermatogonia multiply to self-renew, maintaining a pool of stem cells, and to produce type B spermatogonia, with the latter progressing through mitosis into primary spermatocytes [9].

## 3. Molecular Mechanisms Involved in Radiation-Induced Damage to Spermatogenesis

Ionizing radiation causes damage to all cellular organelles, but the most prominent lesions, relevant to radiotherapy, occur in the DNA molecule, often resulting in cellular death [12]. Figure 2 summarizes the types of radiation-induced damage as well as the molecular mechanisms enabled in sperm cells in response to provoked lesions. DNA damage may happen either by direct ionization of the molecule or more frequently by indirect ionizations caused by free radicals and more specifically by reactive oxygen species (ROS), such as e_aq_, OH° and H°, which are mainly produced by the radiolysis of water [13]. In the context of sperm cell lineage, the generation of ROS directly leads to a variety of DNA lesions, including chromosome deletions, inter-chromatin cross-links and single- and double-strand breaks, with the latter being considered the most lethal lesions for cells [14]. In addition, ROS further potentiate DNA strand breaks through the induction of the apoptotic mediators cytochrome c and caspases 9 and 3 [15]. Regarding the cellular response to radiation-induced DNA damage, apoptotic death comprises the pivotal mechanism by which abnormal spermatogonial cells are depleted from the epithelium [16]. Transformation-related protein p53 (Trp53) (or Tumor protein p53 in humans) is considered a major factor regarding DNA damage–repair and the initiation of apoptotic pathways [17]. Interestingly, there are several studies that support the existence of a unique DNA repair mechanism in spermatogonia and report that, contrary to somatic cells, radiation-induced apoptosis in spermatogonial stem cells is not dependent on Trp53 activation [18,19,20]. However, in contrast with the aforementioned data, more contemporary studies advocate that following radiation-induced DNA damage in spermatogonia, spermatocytes and round spermatids, p53 is indeed recruited, along with other components of the DNA damage–repair and damage-signaling apparatus, including Bcl-2-binding component 3 (BBC3) and tumor protein P53-inducible nuclear protein 1 (Trp53inp1), downstream to the phosphorylation of the histone H2AX [21,22]. The activation of p53 leads to damage-induced apoptosis of the spermatogonia after irradiation with doses as low as 10 mGy [23]. Another signaling cascade that is activated when testicular cells (germinal and non-germinal ones) are exposed to various harmful stimuli, including radiation, is the mitogen-activated protein kinase (MAPK) pathway, which includes the stress-activated protein kinases p38MAPKs and c-Jun N-terminal kinases (JNKs). This pathway not only controls testis development and germ cell maturation and differentiation but it is also implicated in cellular stress response and germ cell apoptosis, being a potential key factor in stress-derived spermatogenic dysregulation and subsequent reductions in semen quality and fertility [24,25]. The inhibition of the p38MAPK pathway in mice has shown promising results in alleviating radiation-induced testicular damage and in enhancing spermatogenesis restoration [26]. Lastly, little progress has been made in determining biomarkers for increased testicular radiosensitivity at the individual level. Research has been focused on the mismatch repair (MMR) genes, whose alterations have been implicated in the etiology of male infertility [27]. Specifically, there is statistically robust evidence that the MutS protein homolog 5 (MSH5) gene variant C85T (Pro29Ser) is associated with an increased risk of male infertility and radiotherapy-induced spermatogenic dysregulation [28,29].

## 4. Research on Animal Models on Radiation-Induced Effects on Spermatogenesis

The first data regarding the impact of radiation exposure on sperm cell production emerged from studies conducted on animals, soon after the discovery of X-rays by Wilhelm Conrad Roentgen [7]. In 1903, the German scientist Albers-Schönberg experimented by irradiating the testes of male guinea pigs and rabbits, observing that despite the preservation of potency, no offspring could be produced by the pairing of females with irradiated male animals. Five years later, H. Frieben focused on potential post-radiation histological changes and showed that exposure of animal testes to radiation led to atrophy of the seminiferous tubules, with degenerated cells in the tubule lumina and complete absence of mitoses in the tubule cell layers [33]. An insight into the differential impact of radiation on different cell lines of testicular tissue was provided by the animal studies by Bergonié and Tribondeau and later confirmed and enriched by other investigators. It was demonstrated that irradiation of animal testes with X-rays caused, proportionate to the dose, selective elimination of the spermatogonia and the inability to further produce sperm cells with subsequent sterility, while at the same time sparing the interstitial cells, Sertoli cells and already produced mature spermatozoa. However, even though the structure and motility of the existing mature spermatozoa appeared to be unaffected, it was noted that these sperm cells were incapable of fertilizing ova. Moreover, despite irradiation, investigators concluded that sexual behavior and ardor remained uninfluenced [33,34].

More contemporary data on animal models have shed light on the dose-dependent manner in which radiation interferes with spermatogenesis, as well as on the underlying mechanisms of disrupted testicular physiology following irradiation. Studies conducted on rodents have shown variable results, mainly due to the use of different species and strains [35]. The restoration of spermatogenesis in mice models is more direct than in humans, with the number of spermatogonia reaching the control levels and sperm production reaching 60% of the control capacity within 2 and 23 weeks, respectively, after reported doses of 6 Gy [36]. When irradiation was escalated to 9 Gy, only a quarter of the seminiferous tubules exhibited sperm cell differentiation and production after 5 weeks [37]. The explanation for the occurrence of atrophic tubules after radiation exposure in mice lies mainly in the depletion of stem spermatogonia from the tubule wall [38]. Rat testicles are more prone to be damaged when irradiated, and the literature on rat models has produced less unanimous data, with the results depending on the strain used [35]. A study comparing the recovery of spermatogenesis after a dose of 5 Gy in seven different rat strains reported rates ranging from no recovery to 98% tubule recovery at 10 weeks [35]. Contrary to mice models, the fact that atrophic tubules generally preserve a number of type A spermatogonia capable of maturing into spermatozoa in irradiated rats indicates that the observed atrophy is attributed not only to relevant stem cell diminution but also primarily to potential dysregulation of spermatogonial differentiation [35,39]. Apart from the total radiation dose, the dose rate is also a determinant of the impact that radiation has on sperm production. In a recent study conducted on mice irradiated with a total dose of 2 Gy, low-dose-rate (~3.4 mGy/h) irradiation of the testes had a more pronounced detrimental effect on spermatogenesis compared to high-dose-rate irradiation (~51 Gy/h), with the authors reporting reduced sperm cell motility, higher abnormality rates and a decline in the total sperm, spermatid and spermatogonia populations in the former group [40]. In contrast, when lower radiation dose rates (0.7 mGy/h) were applied, aberrant sperm cells appeared more abundant in the high-dose-rate group, indicating the potential existence of a cut-off point in the dose rate below which the occurrence of DNA damage and cellular abnormalities is impossible [41].

Lastly, given that non-human primates have a testicular anatomy and physiology that closely resemble those of humans, they have been used in studies investigating the spermatogenic effect of radiation [42,43,44]. Testicular doses of 0.5–4.0 Gy in monkeys resulted in a substantial reduction in sperm count and in an increased rate of abnormal sperm cells in the semen [44], with a reported drop of over 80% in the mean sperm concentration on day 35 and a continuous decline until day 60 after irradiation with 2 Gy [42]. After this dose, the number of spermatogonia was less than 5% of the control level, as noted on day 44, gradually increasing to 70% at day 370, indicating that complete restoration of spermatogenesis is a rather time-consuming process in primates [42,44].

## 5. Research on Humans and Clinical Data on Radiation-Induced Effects on Spermatogenesis

As the Bergonié–Tribondeau principle predicts, human male gonads are some of the most radiosensitive tissues in the body, independent of the patient’s age. Delving into the cellular level, spermatogonia, especially type B, exhibit the most sensitivity, as they constitute the most immature components of the seminiferous tubules [45,46]. Even relatively low radiation doses can lead to testicular damage, with increasing doses affecting more mature cells of the germinal epithelial cell lineage [45]. However, these effects are not only dependent on the total radiation dose but also on the fractionation scheme [47,48]. In addition to the relatively obvious effects of radiation directly targeting the testes, scattered radiation produced by treatment of adjacent structures contributes significantly to testicular damage, even if the gonads are shielded [49,50] (Figure 3). Apart from studies on the spermatogenesis damage provoked by radiation treatment of malignant tumors, either testicular or of nearby anatomical areas, valuable human data on the radiation-induced effects on spermatogenesis and the tolerance dose of the testes have been derived from experiments conducted on volunteer inmates [46,51] and from unfortunate events of accidental radiation exposure, as was the case with the Chernobyl disaster [52,53,54].

### 5.1. Single-Dose Irradiation

Regarding single-dose irradiation, the minimum dose required to cause damage to spermatogonia is as low as 0.1 Gy, and doses higher than 2 Gy and 4 Gy affect spermatocytes and spermatids, respectively, with the latter dose range causing a significantly more abrupt onset of azoospermia, occurring in less than 60 days post-irradiation (Figure 1). Focusing on subsequent sperm cell count abnormalities, the decreases in the number of sperm cells and the time needed for recovery are proportional to the doses applied, with a cut-off of 0.8 Gy determining whether oligospermia or azoospermia occurs. Oligospermia after single doses of up to 0.8–1 Gy requires 9–18 months to resolve, whereas single doses between 2 and 3 Gy and above 4 Gy result in azoospermia, requiring 30 months and 5 years, respectively, to recover [45,51]. As a corresponding clinical example of the aforementioned doses, prophylactic hip irradiation with a single fraction of 7 or 8 Gy for the prevention of heterotopic ossification after traumatic injuries can produce scattered testicular doses ranging from 0.03 to 0.4 Gy (mean dose 0.1 Gy), even in the case of testicular shielding use [55,56]. Finally, there is still controversy on the cut-off single-dose limit capable of causing permanent azoospermia. It has been reported that a single dose exceeding 6 Gy may lead to permanent azoospermia [45]. A large male child cancer survivor study concluded that a testicular dose greater than 7.5 Gy was correlated with male infertility and conception inability (HR, 0.12; 95% CI, −0.02 to 0.64), [57] whereas older studies investigating the effects of total body irradiation and reporting on higher single testicular doses of approximately 10 Gy indicated that only about 15% of irradiated patients managed to restore their sperm count or fertility [58,59].

Sertoli cells are also affected by irradiation of human testes, albeit to a lesser degree compared to germ cell lineage. After irradiation at 1.5 Gy, the number and proliferation rate of Sertoli cells and the expression of Anti-Müllerian Hormone (AMH), a glycoprotein produced by Sertoli cells, were all reduced. The observed reduction in the number of Sertoli cells was more modest compared to that seen for sperm cells, even though both cell lines seem to share the same pro-apoptotic pathways, which rely on the activation of p53 [30].

### 5.2. Accidental Radiation Exposure

A study focusing on the fertility status of eight male workers 3.5 years after exposure to nuclear radiation of 0.22–3.65 Gy in an accident that occurred in the nuclear plants of Oak Ridge reported that at least five out of the eight individuals were sterile within 4 months following exposure. Sterility was maintained even for 30 months in those exposed to higher doses (≥2.36 Gy), with accompanying sperm morphological and motility changes, and a reasonable to good fertility state was regained no sooner than 41 months following exposure [54]. Similar results have been published in studies following the Chernobyl nuclear accident [52,53]. Bartoov et al. included 18 exposed decontamination workers and a corresponding control group for comparison. With a mean radiation exposure dose of 0.09 Gy, the exposed group had statistically significantly fewer motile and progressively motile spermatozoa with more frequent morphological abnormalities. The fertility index used, which was calculated based on quantitative (semen volume and sperm count) and qualitative (percentage motility and normal forms on conventional and electron microscopy) parameters, was substantially lower in the exposed individuals, 39% of whom were characterized as infertile [52].

### 5.3. Direct Irradiation of Testicular Tumours

Many patients diagnosed with seminomas, acute lymphoblastic leukemias, lymphomas and sarcomas or patients requiring total body irradiation receive fractionated radiotherapy directed to or including the testes, with subsequent well-established effects on both exocrine (sperm production) and endocrine testicular function [45,48]. Interestingly, spermatogenic restoration after fractionated radiotherapy is more time-consuming compared to single-dose schemes, an observation described as “the reverse fractionation effect”, and a total testicular dose as low as 2.5 Gy administered in fractions may even lead to permanent azoospermia [47,48]. In a study focusing on long-term fertility in patients with early-stage seminoma managed with radiation treatment with gonadal shielding, 64% successfully sired a natural pregnancy, and despite all patients having partial spermatogenic recovery, 56% presented normal sperm parameters [60]. Long-term survivors of pediatric acute lymphoblastic leukemia who had received testicular irradiation of 24 Gy showed no spermatozoa after semen analysis and a markedly reduced testicular size at a median period of 20 years after the treatment delivery [61]. In a retrospective study, all patients submitted to 9.9 or 13.2 Gy of total body or 6 Gy of thoracoabdominal irradiation combined with cyclophosphamide for subsequent bone marrow transplantation, demonstrated sperm count abnormalities, 85% demonstrated azoospermia and the remaining percentage demonstrated oligospermia at a mean follow-up of 5.6 years. The time for spermatogenic recovery ranged from 4 to 9 years post-transplant [62]. Delving into the radiation-induced effects on sperm DNA integrity, Ståhl et al. analyzed semen from 96 patients diagnosed with testicular cancer and treated with adjuvant radiotherapy with a total dose of 26.2 Gy delivered in 14 fractions. They concluded that DNA strand breaks were significantly more evident during the first 2 years after radiotherapy, and despite the recovery of normozoospermia in 13.5% of irradiated patients after a period of at least 1 year, 38% of them had significant DNA damage (>27%), possibly mitigating their sperm-fertilizing ability [63].

### 5.4. Scattered Radiation from Adjacent Structures

As already mentioned, scattered radiation to the testes, reaching 1.5–3 Gy, is a considerable issue when irradiating adjacent structures, namely the pelvic and thigh areas [48]. Speiser et al. included 10 patients with inverted-Y inguinal irradiation for Hodgkin’s lymphoma and a testicular dose of 1.2–3 Gy delivered in 14–26 fractions. In their first results, all patients presented azoospermia post-treatment, with no evidence of recovery during follow-up periods of over 15 and 40 months in four patients and one patient, respectively [64]. In their updated analysis, patients who had received a dose greater than 1.4 Gy demonstrated persistent azoospermia after a follow-up of 17–43 months, whereas fertility was restored in the two patients with a gonadal dose of 1.2 Gy, possibly revealing the existence of a cut-off dose point below which irreversible testicular damage is perhaps unfeasible [65]. Similar rates of azoospermia after large pelvic-field irradiation or brachytherapy of the prostate were observed by Hahn et al., with 5 out of 11 patients returning to normo- or oligospermia within a follow-up of nearly 9–27 months [66].

## 6. Radiation-Induced Effects on Testicular Endocrine Function

Concerning the other major cellular component of the testes, Leydig cells exhibit far less radiosensitivity compared to germ cell lineage [67]. However, an in vitro study on cultured Leydig cells showed that radiation >6 Gy can lead to cellular dysfunction, mainly affecting the signaling pathways downstream of the LH receptor [68]. In vivo studies have reported that Leydig cells of prepubertal males are more vulnerable to radiation, and their function is preserved up to 20 Gy in this age group, while resistance up to 30 Gy is observed in sexually mature men [48,69,70,71]. Bang et al. compared radiation schemes of 16 and 20 Gy in 8 and 10 fractions, respectively, for patients with in situ testicular carcinoma and concluded that the former is associated with more stable post-treatment testosterone levels. Irradiation with 20 Gy was accompanied by an annual testosterone reduction of 2.4% (*p* = 0.008), more pronounced during the first five years (9% annual decrease) following radiation, and thus a greater percentage of men were in need of androgen substitution therapy in that group (78.6 vs. 48.7% in the 20 and 16 Gy groups, respectively, *p* = 0.03) [72]. In a recently published overview summarizing the existing data regarding the impact of prostate cancer radiotherapy and subsequent incidental testicular irradiation on gonadal function, the authors observed a drop in testosterone levels in all the included trials, irrespective of the treatment technique used (classic three-dimensional conformal radiotherapy or novel intensity-modulated radiation therapy). A dose–response relationship matching incidental testicular irradiation and a reduction in testosterone levels was not established. In most of the studies, the nadir of testosterone levels was reached 3 months after treatment, and despite the complete recovery of testosterone levels in 12 to 18 months post-treatment in the majority, up to 40% of patients were unable to restore their baseline testosterone levels [73].

## 7. Transgenerational Effects of Ionizing Radiation

Following certain unfortunate historical events that resulted in radiation exposure to a considerable part of the general population, as was the case with the Chernobyl nuclear disaster, there was concern regarding the potential hereditary effect of ionizing radiation, with the transmission of radiation-induced germ cell lesions to the offspring of exposed parents [74]. Apart from being a topic of scientific interest, the potential hereditary and transgenerational implications of germ cell irradiation raise significant ethical, legal and social issues, especially in victims of deliberate or accidental radiation exposure from nuclear sources [3]. Although there are several studies based on animal models and experiments that support the existence of such transgenerational effects of ionizing radiation, data on humans remain limited and largely controversial. The extrapolation of results from relative animal studies to human subjects is precarious, as a discrepancy seems to exist between animal experiments and human epidemiological observational studies [3]. Table 1 summarizes data from selected published studies concerning the potential inheritance of radiation-induced DNA alterations and corresponding phenotypic alterations to the progeny of radiation-exposed males.

## 8. Discussion—Implications for Fertility Preservation

Contemporary cancer treatments are constantly becoming more effective and focus primarily on maximizing and targeting their cytotoxic effect towards malignant cells with the aim of curing the disease. However, emphasis should also be placed on treatment safety and the management of potential adverse events, especially irreversible ones, as may be the case with testicular damage and infertility [50,81]. As reported in this review, results across the literature show that even small doses of radiation, which would otherwise be insignificant, may result in severe and sometimes permanent testicular damage and may render such male individuals infertile for a long time period or even indefinitely. It should also be noted that many children and young adults diagnosed with cancer are long-term survivors, with a reported 5-year cancer survival rate of 75% for cancer patients under 15 years old receiving treatment [82]. Over half of these male survivors, who are often childless at the time of diagnosis, will desire to become fathers in the post-treatment future [83].

Taking into consideration the relatively high frequency of gonadal dysfunction as an aftermath of cancer treatment, the induced psychological distress and quality of life impairment correlated with fertility problems and recognizing the need for oncological patients to have access to clinical care regarding their present or future desire for reproduction, [81] the term “oncofertility” was introduced in 2006. It is an umbrella term encompassing the process of discussion with both cancer- as well as fertility-specialized healthcare providers with the goal of raising patient awareness about the possible risks posed by cancer therapy in terms of reproductive health and about the prevention and management of potential complications, such as sexual dysfunction, hormonal abnormalities, delayed puberty, etc. [84,85].

Focusing on fertility preservation in the context of radiation treatment to the male gonads or adjacent structures, shielding of the testes is the first measure taken to minimize radiation damage and preserve reproductive potential. If the risk of testicular damage remains high or in any case of existing concerns about potential post-treatment gonadal dysfunction, fertility preservation strategies should be discussed with the patient in a timely and clear manner before treatment begins. Ethical concerns are raised for patients of childhood age, not only because fertility strategies in prepubertal patients are still largely experimental but also because of difficulties regarding counseling, parent involvement and receiving consent from a patient that has not yet reached adulthood or sexual maturation [81,86].

A multidisciplinary Spanish consensus has published guidelines on the indications for providing fertility preservation in pediatric, adolescent and adult patients with cancer or hematological disease. The consensus proposes offering fertility preservation techniques to all patients with a calculated infertility risk of >50% (intermediate–high- and high-risk) or to patients with low–intermediate gonadotoxic risk (20–50%) and simultaneous comorbidities, like cryptorchidism, monorchidism or previous testicular injury, among others. The aforementioned risk groups include patients receiving total body irradiation for hematopoietic cell transplantation; pelvic radiotherapy with a total dose of >15 Gy or > 10 Gy in pre- and post-pubertal individuals, respectively; total abdominal radiation treatment and craniospinal radiotherapy if the testes are included in the field (0.1–1.2 Gy) [87].

With respect to the currently available fertility preservation strategies, sperm cryopreservation should always comprise the first-line option for males of reproductive age. Contemporary assisted reproduction techniques allow for fertilization with sperm from men with severe oligospermia, and in the context of in vitro fertilization, intracytoplasmic sperm injection with cryopreserved sperm is as effective as fresh sperm use [88]. Regarding the necessary time period before attempting conception, most experts propose a waiting time of 12–24 months post-irradiation [89,90]. However, it should be noted that there are reasonable concerns on the safety of using ejaculated sperm from men who have previously been irradiated and potential detrimental effects on the offspring [91]. Published clinical practice guidelines recommend sperm DNA fragmentation testing after exposure to radiation, as well as after sperm cryopreservation, as both procedures can affect the sperm chromatin structure, which can lead to failed assisted reproductive technology attempts [92]. Finally, the area of fertility preservation techniques in prepubertal boys is still experimental, with cryopreservation of the testicular tissue and subsequent autologous transplantation of spermatogonia being the most auspicious approach. Grafting of testicular tissue samples or in vitro spermatogenesis from spermatogonia or even from primordial stem cells are alternative strategies under research [87,93].

As a last remark, sufficient knowledge about the effect of ionizing radiation on spermatogenesis and relative awareness should always be key concerns, not only in the context of therapeutic irradiation but also in the unfortunate, yet possible, event of unexpected radiation exposure due to nuclear disasters or accidents, as recent human history has shown [94].

## 9. Conclusions

This comprehensive review attempted to highlight the crucial impact of cancer radiotherapy on testicular function, with a particular focus on radiation-induced spermatogenic effects. Both experiments on animal models and data from clinical studies and from accidental radiation exposure have proven the high vulnerability of gonadal function to even relatively low radiation doses, resulting in quantitative and qualitative sperm abnormalities and sometimes endocrine dysfunction, often lasting for long periods or even indefinitely. In an effort to minimize such complications and the induced physiological and emotional distress caused by gonadal dysfunction, all involved healthcare providers should inform patients about and discuss the potential infertility risks and, through the existing fertility preservation strategies, ensure the maintenance of individuals’ reproductive health in the best possible manner. Lastly, even though the potential transgenerational effects of radiation are of great scientific and ethical significance, there is still controversy pertaining to this issue in humans, and future research is required to clarify this issue.

## Figures and Tables

**Figure 1 biomedicines-12-01492-f001:**
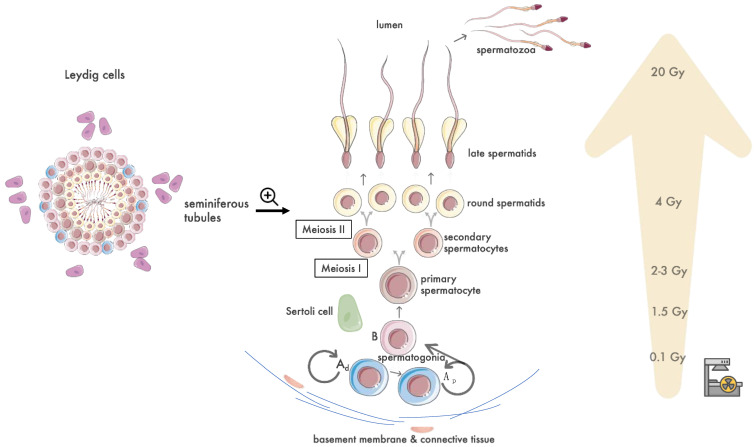
Overview of human spermatogenesis in correlation with the radiation doses required to cause damage to the various cellular components. Spermatogonia exhibit the most radiosensitivity, with damage from doses as little as 0.1 Gy. Spermatocytes are affected by doses ranging between 2 and 3 Gy, whereas spermatids are damaged by doses above 4 Gy. Sertoli cells are also affected by irradiation but to a lesser degree than spermatogonia, with 1.5 Gy causing an observed reduction in their number. Leydig cells are far more radioresistant to up to doses of 20 Gy and 30 Gy in pre- and post-pubertal males, respectively.

**Figure 2 biomedicines-12-01492-f002:**
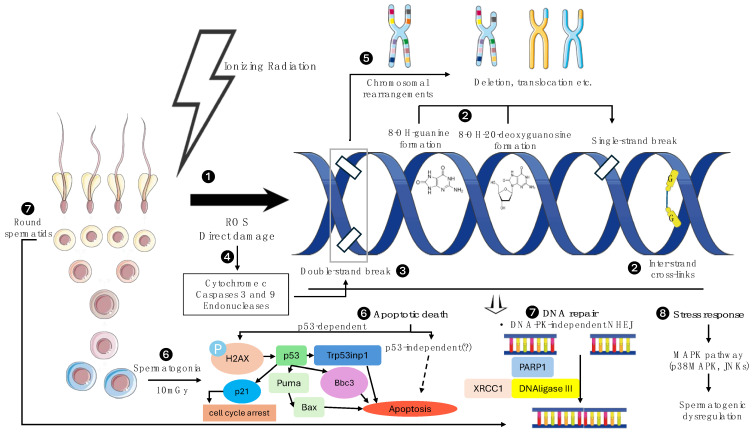
Types of damage and molecular sequelae after sperm cell irradiation. Ionizing radiation, either directly or through ROS production, causes sperm cell DNA damage and ultimately sperm DNA fragmentation (1). This is specifically induced by the formation of inter-strand cross-links and the generation of 8-OH-guanine and 8-OH-20-deoxyguanosine, which can provoke single-strand breaks (2). However, the most severe lesions are double-strand breaks (DSBs) (3), which are an aftermath of either direct radiation impact on DNA molecules or induced by the ROS-mediated activation of cytochrome c, caspases 3 and 9 and endonucleases (4). Unrepaired double-strand breaks may lead to chromosomal rearrangements, including deletions, translocations, etc. (5). The aforementioned lesions in spermatogonia, spermatocytes and spermatids can activate p53-dependent and possibly p53-independent apoptotic pathways, with a cut-off dose as low as 10 mGy for apoptosis of spermatogonia (6). Alternatively, sperm cells can repair radiation-induced DNA damage. Round spermatids employ DNA-PK independent non-homologous end joining, an alternative back-up pathway, to repair DSBs (7). Lastly, ionizing radiation stimulates stress-response-related cascades, such as the MAPK pathway, which are involved in spermatogenic dysregulation (8). Bbbc, Bcl-2-binding component 3; DNA-PK, DNA-dependent protein kinase; G, guanine; H2AX, H2A histone family member X; JNKs, c-Jun N-terminal kinases; MAPK, mitogen-activated protein kinases; NHEJ, non-homologous end joining; P, phosphorylation; PARP1, poly (ADP-ribose) polymerase 1, ROS, reactive oxygen species; Trp53inp1, tumor protein p53-inducible nuclear protein 1; XRCC1, X-ray repair cross-complementing protein 1 [3,30,31,32].

**Figure 3 biomedicines-12-01492-f003:**
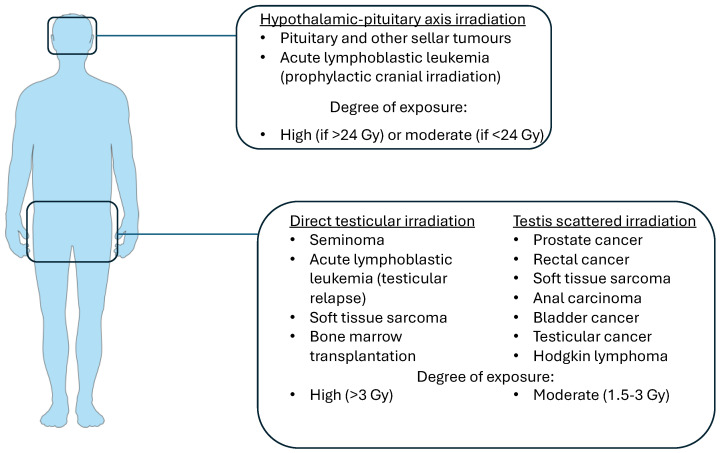
Diagram of diseases associated with potential fertility complications in the context of radiotherapy (adjusted from Felice FD et al. [48]).

**Table 1 biomedicines-12-01492-t001:** Selected studies on transgenerational effects of ionizing radiation.

Authors	Year	Type of Study	Source of Radiation	Findings
Gardner et al. [75]	1990	Case–control, humans	Sellafield nuclear plant	Relative risks of leukemia and Hodgkin’s lymphoma were statistically significantly higher in children born from fathers with high radiation dose recordings before their child’s conception
Dubrova et al. [76]	2000	Laboratory study, mice	0.5 Gy fission neutrons	The indirect effect of radiation extends to the germline of unexposed first-generation offspring of irradiated male mice
Izumi et al. [77]	2003	Cohort study, humans	Atomic bombs in Hiroshima and Nagasaki	No difference in cancer incidence between subjects with exposed parents and reference subjects
Winther et al. [78]	2012	Case–cohort study, humans	Radiotherapy and/or chemotherapy for the treatment of cancer	No statistically significant difference in the risk of genetic disease between children of irradiated parents and non-irradiated ones
Yeager et al. [79]	2021	Observational study, humans	Chernobyl nuclear accident (mean = 365 mGy, range = 0–4080 mGy)	No increase in the rates, distributions or types of germline de novo mutations compared with previous studies
Li et al. [80]	2024	Laboratory study, mice	Whole body acute gamma irradiation at 6.4 Gy	Male mice could have healthy offspring after post-irradiation recovery of fertility. The reproductive, metabolic and neurodevelopmental health of offspring born to irradiated undifferentiated spermatogonia were comparable to those of controls.

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
