# Peer review of "Radiotherapy and Testicular Function: A Comprehensive Review of the Radiation-Induced Effects with an Emphasis on Spermatogenesis"

_biomedicines, 2024, doi:10.3390/biomedicines12071492_

Round 1

Reviewer 1 Report

Comments and Suggestions for Authors

This review manuscript focused on the testicular functions after radiotherapy. It would be interesting to recall the hereditary effect of ionizing radiation, although the doubling dose of human related hereditary effect has been reported to be high.  Given the advanced IGRT or IMRT, the radiation energy should be able to avoid high dose exposure to gonad except necessary. Therefore, low dose radiation expoure could be an issue, but should focus on Mandalian disease or disordered chromosome number. The signifance of radiation caused damage on spermatogenesis should be considered and addressed well. Specific comments include:

1. Figure 1 should point out the types of damage of different dosages on different stages of spermatogenesis. 

2. Is there any new findings on hereditary effect? Please summarize it with a Table.

3. What types of diseases required radiotherapy that may influence the goand function and spematogenesis? A comprehensive diagram would be nice for readers.

4. A perspective on future direction of study on spermatogenesis and hereditary, especially for the events of nuclear wars and explosion of nuclear plant should also be included. Compared to radiotherapy, these nuclear events may occur unexpectedly and very possible.

Reviewer 2 Report

Comments and Suggestions for Authors

1. It is mentioned that endocrine function is affected, including male sexual function. Do the authors have relevant data analysis on the effect of radiotherapy on male erection?

2. Why do spermatogonia and Sertoli cells respond differently to radiotherapy?

Round 2

Reviewer 1 Report

Comments and Suggestions for Authors

The authors have well addressed my last comments point-by-point. The revised version is great and should be appropriate for publication. 

Author Response

Below you will find a list of your comments accompanied by a point by point response to each one, including the corresponding changes made by us.

Comment:

“In Figure 2, spacing errors are evident. For instance, the first letters of some words are obscured by adjacent numbers.”

Action:

Quality and spacing errors of Figure 2 (provoked by the conversion of the “Word” document to the PDF version) were corrected.

Comment:

the manuscript would benefit from professional editing to enhance readability and address grammatical errors, including run-on sentences. For instance, lines 215-218. Lines 21-26 and 256-261 contain repetitive coordinating conjunctions. Both use ‘but’ twice instead of employing synonyms.”

Action:

Done. We replaced the conjunction word “but” in line 27 with the word “however” and instead of repeating “but” in lines 255-261, we used the conjunction words “albeit” (line 255) and “even though” (line 259). Moreover, we changed the structure of the sentence in lines 30-32 to make it simpler and to avoid overuse of the word “regarding”. We also corrected other minor errors, including double spacing and punctuation mistakes.   

            We hope that the applied corrections contribute to the improvement of the manuscript text and that they meet the standards of the journal.

            Thank you for your valuable contribution and your time.

Sincerely,

Ntoumas Georgios-Nikiforos (on behalf of all contributing authors).